# Thrombotic Alterations under Perinatal Hypoxic Conditions: HIF and Other Hypoxic Markers

**DOI:** 10.3390/ijms241914541

**Published:** 2023-09-26

**Authors:** Alejandro Berna-Erro, María Purificacion Granados, Juan Antonio Rosado, Pedro Cosme Redondo

**Affiliations:** 1Department of Physiology (Phycell), University of Extremadura, Avd de la Universidad s/n, 10003 Caceres, Spain; alejandrobe@unex.es (A.B.-E.); pcr@unex.es (P.C.R.); 2Pharmacy Unit of Extremadura County Health Service, Health Center of Talayuela, 10310 Talayuela, Spain; mpc77@gmail.com

**Keywords:** Hypoxia, HIF-1, calcium, platelets, neonates

## Abstract

Hypoxia is considered to be a stressful physiological condition, which may occur during labor and the later stages of pregnancy as a result of, among other reasons, an aged placenta. Therefore, when gestation or labor is prolonged, low oxygen supply to the tissues may last for minutes, and newborns may present breathing problems and may require resuscitation maneuvers. As a result, poor oxygen supply to tissues and to circulating cells may last for longer periods of time, leading to life-threatening conditions. In contrast to the well-known platelet activation that occurs after reperfusion of the tissues due to an ischemia/reperfusion episode, platelet alterations in response to reduced oxygen exposition following labor have been less frequently investigated. Newborns overcome temporal hypoxic conditions by changing their organ functions or by adaptation of the intracellular molecular pathways. In the present review, we aim to analyze the main platelet modifications that appear at the protein level during hypoxia in order to highlight new platelet markers linked to complications arising from temporal hypoxic conditions during labor. Thus, we demonstrate that hypoxia modifies the expression and activity of hypoxic-response proteins (HRPs), including hypoxia-induced factor (HIF-1), endoplasmic reticulum oxidase 1 (Ero1), and carbonic anhydrase (CIX). Finally, we provide updates on research related to the regulation of platelet function due to HRP activation, as well as the role of HRPs in intracellular Ca^2+^ homeostasis.

## 1. Placental Alteration Leading to Fetal Hypoxemia 

The specialization of throphoblast cells allows for the generation of a complex functional tissue that has two main roles during the development of the fetus: (1) oxygen supply; and (2) the generation of a physical barrier that controls the interchange of particles and cells between the mother and the fetus. Throphoblastic cells interact with two vascular beds, avoiding direct contact between maternal and neonatal blood: the uterus vascular bed and extraembryonic mesenchyme of the allantois [1]. Uterine-linked trophoblast cells are responsible for supplying maternal blood to placenta, while trophoblast cells from the allantoic vasculature efficiently extract nutrients that end in the fetal vasculature [1]. Low oxygen supply to the embryo has been demonstrated before the implantation phase, and this stressful situation drastically changes upon placenta formation. Interestingly, at up to 12 weeks of gestation, a difference in the partial pressure of oxygen between the placenta (17.9 mm Hg) and endometrium (39.6 mm Hg) can be seen, providing evidence of hypoxemia during this stage of embryo development [2]. 

Placental alterations due to hypertension or preeclampsia are present in 3–7% of pregnant women worldwide. Often, defective trophoblastic invasion and an impaired development of the uterine spiral arteries lead to abnormal placentation, the main cause of hypoxia/reoxygenation phenomena; further, it may result in fluctuations of the gradient of oxygen, altered antioxidant capacity, placenta oxidative stress; and a reduction in nitric oxide (NO) [3]. On the other hand, low oxygen supply to the fetus may occur due to early senescence of the placenta. Senescence of the placenta is characterized by the increased expression of p53, p21, and p16 transcriptional factors, as well as premature senescence of the extravillous trophoblasts. The latter is associated with abnormal placental development, that in turn leads to blood hypoperfusion and, subsequently, develops into fetal growth restriction, preterm birth, and stillbirth [4].

## 2. Hypoxia during Labor

Oxygen saturation (SpO_2_) blood tests determine the percentage of hemoglobin bound to oxygen. It is an indirect indicator of tissue oxygenation. The average arterial SpO_2_ value usually found in healthy adults is around 98%, and this indirectly indicates an adequate oxygen supply to the whole organism. The threshold considered as “below normal” oxygen saturation (hypoxemia) is when SpO_2_ values are below 90% [5]. Hypoxemia is not a dangerous condition as long as tissues are adequately prefunded and oxygenated. Conversely, severe hypoxemia can lead to defective tissue oxygenation or to hypoxia that comprises tissue damage and leads to subsequent deleterious effects in physiology and metabolism [6].

### 2.1. Risk of Perinatal Hypoxemia and Hypoxia during Gestation

The fetus must face hypoxemia during labor, since oxygen saturation in arterial blood drops considerably, lasting several minutes after parturition [7]. The fetus grows in a hypoxemic environment in utero that, surprisingly, seems to be optimal for its development. For instance, SpO_2_ values of around 70% are frequently found in the umbilical vein as compared with that of 98% found in mothers [8]. Thus, the fetus should have a particular metabolism that is adapted to mild hypoxemic conditions; for instance, fetuses express increased levels of fetal hemoglobin, which is much more efficient at capturing oxygen than adult hemoglobin [9]. Immediately before parturition, fetal oxygen saturation drops drastically after each uterine contraction [10]. Next, the fetus must face an additional downward trend in oxygen saturation and a marked transient hypoxemia during delivery, so that the mean preductal SpO_2_ decreases to 50–73% around 10 min before parturition, re-marking the hypoxic temporal conditions. However, SpO_2_ values quickly increase up to 90% after 5 min upon the start of breathing, thereafter achieving the normal values (95%) found in adults after 12 min, or up to 97.6% during the first week after birth [7,11]. Each labor varies in time and in the level of fetal stress. These can be prolonged under certain situations or complications, further increasing the length and degree of neonatal hypoxemia. For instance, SpO_2_ values in arterial blood decrease by 3% during cesarean delivery as compared to vaginal parturition [12]. This transient hypoxemia during normal labor does not pose a threat, but newborns must adapt themselves quickly and increase oxygen saturation in the blood by oxygen intake from a new, rich oxygen atmosphere. If the transition fails, neonates suffer prolonged hypoxemia that can develop into life-threating asphyxia and hypoxia, a prevailing complication in labor as compared to other stages of life [13]. The appearance of neonatal hypoxia as a consequence of prolonged hypoxemia indicates the existence of a transient adaptative mechanism against neonatal damage, since newborns are not indefinitely “protected” from prolonged hypoxemia. Complications that can lead to fetal and neonatal hypoxia include maternal heart, lung and kidney diseases, but also decreased maternal breathing from anesthesia, anemia, maternal diabetes, or low maternal blood pressure. Additionally, pathologic intrauterine hypoxia also arises as a consequence of maternal hypertension and preeclampsia; fetal intrauterine growth restriction; asphyxia due to compression of the umbilical cord; premature birth; traumatic delivery; intravascular volume contraction after birth; immature mucosal and skin barriers; the failure of lung expansion; leftover fluid in the lungs; and meconium aspiration [13,14]. 

### 2.2. Risk of Hypoxia after Birth

Two out of every thousand newborns suffer from asphyxia after birth in developed countries, a number that rises tenfold in the absence of healthcare in non-developed countries [15]. A common disorder that triggers neonatal hypoxia is hyaline membrane disease, or neonatal respiratory distress syndrome (NRDS). NRDS is a failure in pulmonary surfactant maturation that impedes adequate lung and alveoli expansion within 24 h after birth, hindering correct pulmonary contraction and normal breathing. The prevalence of NRDS is higher in preterm newborns because their lungs are not ready to produce enough surfactants. Inflammation, due to meconium aspiration or previous obstructive processes, also leads to NRDS, worsening hypoxemia conditions during labor [16]. Uncontrolled subsequent hypoxia as a consequence of asphyxia, together with increased inflammation derived from NRDS, often leads to additional complications, such as pulmonary hypertension in newborns (PHN), which increases hypoxia and further damages tissue mostly localized in the pulmonary vascular tree. Uncontrolled severe respiratory distress syndrome can impede complete lung development and it can progress to permanent damage over time, frequently evolving towards bronchopulmonary dysplasia (BPD), among other chronic breathing problems [17]. Common complications of neonatal hypoxic disease include sepsis, asthma, defective vision, or a delay in learning body movements. Serious complications include bleeding or thrombotic ischemia in the brain, with further consequences such as delayed cognitive development leading to learning or behavioral disabilities, and even cerebral palsy, multiorgan failure, and neonatal death. Among others, disorders in neonatal blood coagulation, usually referred to as thrombotic and hemostatic disorders, are the most common complications associated with neonatal hypoxic pathologies. Hypoxia/asphyxia causes altered coagulation in the immature neonatal hemostatic system, greatly complicating the course of the disease, and increasing the risk of perinatal death. Thrombotic disorders may appear localized, for instance, in the pulmonary vasculature, or scattered all over the vascular tree. Further, respiratory distress syndrome leads to endothelial activation and decreased numbers of circulating platelets (thrombocytopenia) due to increased platelet adhesion [16]. Untreated hypoxia often leads to intraventricular hemorrhage, a common cause of death in preterm infants. The complete mechanism underlying this pathology is not clear, but an immature neonatal hemostatic system contributes to the appearance of the disease [18]. 

### 2.3. Hypoxic-Evoked Thrombotic Alterations during Labor 

Platelets are the main cellular components of the hemostatic system. Platelets rise from megakaryocytic fragmentation within the bone marrow in response to thrombopoietin [19]. Thrombopoietin, combined with interleukins and others, induces megakaryocyte transformation into giant multinuclear cells, leading to cell cytoplasmic fragmentation and, finally, evoking the generation of pro-platelets that subsequently mature into platelets due to fragmentation into pulmonary vasculature [19]. Platelets participate in the coagulation cascade that comprises a wide range of proteins generated in the liver and released to the blood stream, which act as agonists (pro-thrombotic) or inhibitors (anti-thrombotic) of blood coagulation. Platelets work together with the coagulation cascade to stop excessive blood loss during traumatic hemorrhage (hemostasis). Bleeding arrest is achieved by plugging broken vessels with a clot of aggregated platelets and blood cells (thrombus). Pathological uncontrolled thrombi formation may lead to obstructive complications (thrombosis), leading to vessel occlusion. Severe vessel occlusion may compromise blood supply and organ function. Platelets also participate in inflammation and immunity [20].

Increasing evidence indicates the maturation of fetal hemostatic system in childhood [21]. Thus, studies in murine models suggest the existence of megakaryocytic and hematopoietic transcriptional networks controlling developmental changes in platelets [22]. This implies significant differences between newborns and adults. For instance, neonatal megakaryocytes are smaller but hyperproliferative compared to those of adults [23]. Neonatal platelets are less reactive or hyporeactive in response to principle physiologic agonists such as ADP, thrombin or collagen [24,25,26]. Key surface proteins involved in platelet activation are also decreased [27]. These differences are much more evident in preterm newborns, who present longer bleeding times [28]. Mechanisms underlying neonatal hyporeactivity are not completely understood, but the proposed hypotheses are decreased receptor expression, changes in signaling pathways. or decreased intracellular Ca^2+^ mobilization [29]. However, decreased neonatal platelet activity is counterbalanced by a more robust response by the coagulation cascade [30], increasing, for instance, the expression and activity of the pro-thrombotic von Willebrand factor (vWF) [31].

Conversely, evidence suggests that the concentrations of some pro-thrombotic factors of the coagulation cascade are decreased, although this requires further investigation [32]. Interestingly, anti-thrombotic factors of the coagulation cascade, such as protein C, protein S, antithrombin III, heparin cofactor II and plasminogen, are also decreased, showing less inhibitory and fibrinolytic activity, thereby compensating the minor presence of pro-thrombotic factors [33]. This decrease in anti-thrombotic factors reinforces pro-thrombotic activity, indicating that neonatal hemostasis mostly relies on the coagulation cascade, as compared to adults. The consequence of the reinforced role of the coagulation cascade in newborns is an increased prevalence of thromboembolic complications when the coagulation cascade is pathologically altered. Additionally, increasing evidence suggest that therapies developed for adults but given to newborns should be adjusted. In this sense, platelet transfusion often solves the problem of neonatal thrombocytopenia, but this procedure is not without risks and it is under evaluation [34]. Transfusion of adult platelet plasma to thrombocytopenic newborns shortens aggregation rates and increases the risk of thrombosis due to the reinforced neonatal coagulation cascade [35]. Additionally, thrombocytopenic neonates must face other adverse effects associated with transfusion [36]. 

The incidence of thromboembolic complications during labor is higher than in other stages of life because of the traumatic nature of this event. One fourth of neonates, for instance, suffer from thrombocytopenia. Severe thrombocytopenia often leads to prolonged bleeding times, increasing the incidence of hemorrhage [37]. Neonatal thrombocytopenia may result from complications during fetal development, during labor or immediately after parturition, hence representing a serious risk factor for mortality in newborns. As we have noted, the fetus grows normally in a hypoxemic environment. However, severe intrauterine hypoxia as a consequence of maternal pathologies may lead to neonatal thrombocytopenia. In this case, the cause of thrombocytopenia arises from impaired platelet production by defective megakaryocytic function [37]. Megakaryopoiesis significantly increases in asphyxiated infants, but decreases as platelet counts reach normal numbers, indicating a possible compensation mechanism for low platelet production [38].

## 3. Altered Hemostasis during Exposition to Perinatal Hypoxia

### 3.1. Changes in the Coagulation System

Asphyxiated newborns, as a consequence of complications after birth, develop mild or severe thrombosis among other disorders, followed by secondary thrombocytopenia, which drastically increases the risk of morbidity [37]. The molecular mechanism is under investigation, but the relevant role of the coagulation cascade in their immature hemostatic systems seems to sensitize them to these coagulation disorders. Together with an associated inflammation, hypoxia damages and activates the endothelial cells of the vascular wall. These cells react to the injury, increasing the production of the vasoconstrictor endothelin-1 (ET-1), pro-thrombotic reactive oxygen species (ROS), thromboxane A_2_ (TxA_2_) and tissue factor, but conversely decreasing the production of anti-thrombotic and vasodilators such as nitric oxide (NO) and prostacyclin (PGI_2_). Antithrombotic factors of the coagulation cascade, such as antithrombin III, protein C, and protein S, also decrease dramatically during asphyxia, reducing the fibrinolytic activity necessary for the prevention of thrombus formation and clearance (thrombolysis) [39]. Activated endothelial cells, together with increased pro-thrombotic ROS and TxA_2_, activate circulating platelets in turn [40], promoting their adhesion to the vessel wall and generating thrombocytopenia due to this excessive platelet consumption. Decreased anti-thrombotic factors cannot counter-balance this process. 

### 3.2. Changes in Platelet Reactivity

In the absence of neonatal evidence, studies in adults suggest that hypoxic stress increases the reactivity of platelets by changing their protein expression profile [41]. For instance, stressed platelets express hypoxia-inducible factor (HIF)-2, which increases the expression and release of pro-thrombotic factors such as plasminogen-activator inhibitor-1 (PAI-1). This increase in activation and granule secretion is mainly achieved by calcium (Ca^2+^) entry through tstore-operated calcium entry (SOCE) into the platelets [42,43,44]. Rodent models also point to an increase in cyclooxygenase-2 (COX-2) in platelets, as well as its main product TxA_2_, which greatly amplifies their activation and aggregation [43,45]. As a consequence, severe hypoxia often leads to excessive platelet activation and scattered adhesion to the endothelium of the vascular tree, referred to as disseminated intravascular coagulation (DIC). The prevalence of DIC is particularly high in neonates, which may result in life-threatening conditions. Commonly targeted organs during the development of DIC are the lungs; liver; and kidneys, frequently as a result of renal venous thrombosis [46]; as well as the brain, due to cerebral sinovenous thrombosis, which increases the risk of neonatal encephalopathy [47]. The result is that 50% of neonates who develop DIC finally die [48].

### 3.3. Complications in Newborns as a Consequence of Hypoxia

Moreover, excessive consumption of platelets during DIC results in thrombocytopenia, increasing the risk of excessive bleeding and hemorrhage [49]. In summary, 7.1% of neonates suffering from respiratory distress syndrome, a common cause of asphyxia, develop DIC, 50% develop moderate to severe thrombocytopenia, and 14.3% develop hemorrhagic episodes [50]. 

On the other hand, released factors by the activated endothelium as a consequence of asphyxia also act as vasodilators and vasoconstrictors, and their imbalance promotes the appearance of pulmonary hypertension of the newborn (PHN) as a result of increased vasoconstriction in the pulmonary vessels. Vasoconstriction leads to restricted blood flow, triggering hypoxia or worsening an initial hypoxic condition. PHN occurs in 2 per 1000 newborns, normally as a complication of a previous existing disease. For instance, 10% of newborns with failed cardiopulmonary transition develop PHN. Other causes favoring PHN are maternal drug intake, asphyxia, meconium aspiration syndrome, respiratory distress syndrome, or pneumonia. PHN is resolved in most cases, but 5% of newborns develop severe persistent PHN (PPHN), an important risk factor for mortality. Twenty-five percent of survivors develop neurological defects. PHN is also associated with thrombocytopenia, among other complications [51,52]. While platelet activation has been reported in adults with pulmonary hypertension, it has not been explored in neonates. Studies in animal models also suggest that PPHN itself increases platelet activation [53]. Therefore, the increased TxA_2_ generated by endothelial cells and adhered platelets [40], together with ET-1, stimulates contraction of the smooth muscle cells (SMCs), which are located in the vessel wall. Animal models suggest that another secreted factor, such as serotonin, released by the activated platelets might contribute to vasoconstriction and PHN [53]. 

The release of these platelet agonists are mediated by intracellular Ca^2+^ mobilization and Ca^2+^ entry through SOCE [42]. The decrease in NO and PGI_2_ cannot counter-balance vasoconstriction and, therefore, vasoconstrictors, such as thromboxane and endothelin-1, activate receptors coupled to G-proteins (Gq/11R) linked with, among others, an increased inositol triphosphate (IP_3_) and inhibition of the production of cyclic guanosine monophosphate (cGMP) in neonatal SMCs, and, subsequently, evoking enhanced SOCE activation. As result, hypoxia promotes contraction of SMCs and the maintenance of PHN [54,55]. Thus, PHN contributes to the maintenance of hypoxia among other cardiovascular alterations, aggravating the condition and increasing the risk of mortality. Increased fibrin/fibrinogen deposition in the pulmonary vasculature under hypoxia, and the absence of fibrinolytic activity, amplifies this positive feedback loop, and triggers further thrombosis in the lungs [56]. PPHN and hypoxia also induce pulmonary vascular remodeling through increased SOCE, by proliferation in SMCs, and by the expression of hypoxia-sensitive genes triggered by the master hypoxia regulator, HIF-1, thereby leading to pulmonary arterial wall-thickening and further prolonging the pulmonary hypoxia [55]. Neonatal hypoxemia and asphyxia are often treated with extracorporeal oxygenation or by hypothermia to prevent severe brain damage. Unfortunately, a common complication of these treatments is an increased risk of thrombosis, which is solved by anticoagulant administration [57]. For instance, a swine model of asphyxia revealed that therapeutic re-oxygenation triggers unwanted transient platelet activation, eliciting deranged coagulation [58]. The decreased temperature during therapeutic hypothermia slows the hemostatic system down even further, increasing the risk of uncontrolled bleeding and hemorrhage [59].

## 4. Cell Markers of Hypoxia

### 4.1. HIF-1 Is the Keyhole during Hypoxia

Among other proteins, we aim to explore here the most relevant cellular markers of hypoxia, being HIF-1 identified in most tissues investigated under hypoxic conditions. Hypoxia inducible factor 1 is a large protein (HIF-1; 92 kDa) formed by the association of two subunits, alpha and beta. Codified by a gene located at the chromosome 14 (14q23.2) and, despite that it also responds to endoplasmic reticulum stress, both the expression and function of HIF-1 drastically increases in response to hypoxic conditions; therefore, it has been highlighted as the main signaling-regulator during hypoxia. Although the HIF-1β subunit is very stable, independently of the hypoxic conditions, HIF-1α mRNA is constantly expressed by several mammalian cells [60]. However, under normal oxygenation (normoxia), HIF-1α is highly unstable at the protein level. Conversely, under hypoxic conditions, this subunit becomes very stable [60]. As a transcription factor, the former HIF-1 complex associates to DNA sequence 5′-TACGTG-3′ of its targeted genes, which requires a previous interaction with the HIF-1 partner Arnt (DNA-binding factor) that in turn stabilizes the HIF-1α subunit [61].

Other members of the HIF family are HIF-2α and HIF-3α [62] and, despite all of them sharing similarities, it has been shown that they do not present redundant functions. All of them regulate gene transcription in different cell types in response to hypoxia [63]. In fact, under certain circumstances, HIF-1α differently controls to HIF-2α. For instance, in clear cell renal carcinoma, HIF-2α promotes cell survival, while HIF-1α activation leads to cell death [63]. In line with this observation, the hematopoietic growth factor, erythropoietin (EPO), was demonstrated to be upregulated by HIF-2α under hypoxic conditions in an in vitro model of hypoxia–ischemia [64]. Conversely, EPO was shown to downregulate HIF-1α through promoting the activity of the prolyl hydroxylase domain 2 (PHD-2) and MMP-9 [64]. On the other hand, the role of HIF-3α has not yet been well described, but it is known that transcription of the HIF-3α locus results in at least five splice variants [65]. In samples collected from patients suffering hepatic carcinoma, HIF-3α was shown to be upregulated in 46% of the samples; nonetheless, it was also observed to be downregulated in another 42% of cases [66]. On the other hand, the expression of HIF-3α directly correlated with HIF-2α in these samples, but not with HIF-1α. Therefore, considering that HIF-3α only presents a transactivation domain (TADs) that is sensitive to hypoxia, in contraposition with HIF-1α and HIF-2α that contain 2 TADs [67], it has been concluded that HIF-3α may be a hypoxic gene that can be targeted by HIF-1α or HIF-2α. Interestingly, in hepatocellular carcinoma, the overexpression of HIF-2α promoted HIF-3α expression, while HIF-1α remained unmodified [66]. In contrast, HIF-3α was largely found to co-localize with HIF-1α in human renal carcinoma cells, but the regulation of HIF-3α remains elusive. It has been clearly demonstrated in the literature that HIF-3α function, once it becomes active, downregulates the function of HIF-1α and HIF-2α [68]. A recent study indicates that HIF-3α can be regulated by methylation in two gene regions, and that methylation in the second region could be linked with pre-eclampsia, as demonstrated by analyzing HIF-3α in mononuclear cells isolated from umbilical cord blood [69]; however, this study was largely descriptive and the authors did not delve into the molecular mechanism behind the altered methylation of the HIF-3α gene. 

### 4.2. Regulation of HIF-1 Function

Regarding the regulation of HIF-1α, several proteins have been proposed, such as MUC-1, Pin1, and the tumor suppressor gene von Hippel–Lindau protein (VHL), among others, as presented in Figure 1 [70]. In normoxia, the presence of oxygen favors HIF-1α inactivation due to hydroxylation at the two proline residues (Pro402 and Pro564) that belong to the alpha subunit; as a result, HIF-1α interacts with VHL and the E3 ubiquitin ligase complex to favor the degradation of HIF-1α [70]. Furthermore, hydroxylation at the asparagine residue within the carboxy-terminal domain blocks interaction with the coactivator of HIF-1α, the p300 [71]. The proposed mechanism works as follows: prolyl hydroxylase (PHD) detects the O_2_ concentration at cellular level and then evokes the hydroxylation at the ODD domain of HIF-1α [72]. In hypoxia, since the hydroxylation enzymes, PHD1-3, require O_2_ and α-ketoglutarate as substrates, the privation of O_2_ should be able to reduce hydroxylation at the prolyl residues and, therefore, impair the ubiquitination and degradation of HIF-1α. The accumulation of HIF-1α subunits and their binding to HIF-1β resembles the active HIF-1α factor that associates with the Arnt cofactor and facilitates the recognition and association with the hypoxia-response elements (HER) 3’-flanking regions of the targeted genes, favoring its translation [70,73]. 

HIF-1α could be degraded by SUMOylation, proteasomal degradation machinery and/or chaperone mediated autophagy [70]. Therefore, regulation of HIF-1α relies on a delicate balance between the activity of those proteins that favors its degradation and those that avoid it. Among others, proteins like HEXIM1, MCM7, OS9, PLD1, RHOBTB3, RUNX3, SIRT2, SPRY2, SSAT2 and WWOX promote the normoxia-dependent and PHD/VHL-dependent degradation of HIF-1. Conversely, ATP6V0C, NQ01, OTUD7B, RSUME, RUNX2, SENP1, UCHL1, USP8 and ISP30 interact with HIF-1α to avoid its O_2_-dependent degradation [71]. In addition, O_2_-independent degradation of HIF-1α was reported to occur upon its interaction with the following proteins: BHLHE41, CDK2, CHIP/HSP70, CHIP/HSC70/LAMP2A, HAF, P53, PIASY, RACK1 SIRT1 SIRT7, SSAT1, and TAp73. Meanwhile, HIF-1α interaction with BCL2, CDK1, HSP90, MUC1 SEPT9 and TRAF6 seems to avoid O_2_-independent degradation [71,72]. In addition, in silico analysis revealed that many other proteins may interact with HIF-1α, but the relevance of these interactions in cell physiology are not defined yet [73].

The HIF-1 targeted genes list includes: TIMP-1 (matrix and barriers reorganizing function) [74], CD18 (inflammation) [75]; EPO (erythropoietin) [76]; TF and TFRC (iron metabolism) [77]; the VEGF signaling pathway that regulates angiogenesis (including VEGF, EGF, Flt-1, PAI-1, ANGPT, Tie-2 and TIMP-1) [78]; EDN1, iNOs, eNOS, HMOX1 and ANP that are involved in vascular tone [79]; Glut1, which promotes anaerobic metabolism by increasing glycolysis; PDK-1, an inhibitor of tricarboxylic acid cycle metabolism; hexokinase, PFKL, GAPDH, ALDOA, ENO1, PGK1, PFK3 and LDHA that are involved in the promotion of the anaerobic metabolism [80]; and Bcl-2 and p21/p27, as regulators of proliferation and apoptosis [81]. Finally, a novel actor has been described downstream of HIF-1, stanniocalcin-2, whose role remains largely unknown but its expression in cancer cells was upregulated and it seems to protect cells from apoptosis [82]. In addition, stanniocalcin-2 was demonstrated to participate in the regulation of Ca^2+^ homeostasis in human platelets by acting as an antagonist of extracellular non-capacitative calcium entry driven by Orai3 [83]. In line with this observation, intermittent hypoxia favors ROS production that activates PLCγ, generating IP_3_ that mobilizes Ca^2+^ and evokes the activation of calmodulin kinase, CamK [84]. CamK activation phosphorylates and recruits p300/CBP to the HIF-1α/β complex, favoring the activation of the complex and, subsequently, favoring the translation of the downstream genes [84,85]. The expression of the STC2 gene may require the recruitment of p300 and HDAC7, according to the researchers’ observations [84]. Similarly, the STC2 paralogue, STC1, was reported to also be upregulated by hypoxia in an HIF-1 dependent mechanism in rat alveolar type II cells [86]. Finally, the activity of HIF-1α was reported to be blocked by drugs like PX-478 2HCl, which was used in in vitro studies to induce apoptosis in tumoral cells, and is in phase 1 of a clinical trial with the reference number, NCT00522652; physicians are now testing its efficacy in treating advanced solid tumors and lymphoma.

### 4.3. Alternative Hypoxic-Sensitive Signaling Pathways to HIF-1

The activation of alternative signaling pathways, independent to HIF-1, have been described in cells under hypoxic conditions (see Table 1, which represents those actually expressed in human platelets). In the following section, we describe relevant alternative markers and second messengers involved in cell survival under hypoxic conditions, including the Carbonic anhydrase IX (CAIX), ERO1a, KDM4B, KDM3A, PDK1, GLUT1, osteopontin, and BNIP-3.

Carbonic anhydrase family (CA) groups contain fifteen members, the most relevant of which are the CAIX and CAXII members. As zinc metalloenzymes, they regulate pH within the cells and in their surrounding medium by transforming the CO_2_ to bicarbonate [92]. Tumor cells maintain an acidic pH in the surrounding microenvironment due to CAIX activity, thus avoiding the activation of a hypoxia-related cascade [92]. However, in most of the tumor samples investigated, CAIX seems to be upregulated by HIF-1; CAIX(−)/HIF-1α (+) and CAIX (+)/HIF-1α (–) can sometimes be detected in cervix cancer samples [93]. Interestingly, CAIX overexpression was also observed in cells grown in high densities, where its activation seems to relay in PI3K activity instead of on HIF-1 activation [93].

Endoplasmic reticulum oxidase 1 (ERO1) is a glycosylated flavonase mainly located at the mitochondria-associated membrane (MAMs), where it regulates the ER redox and Ca^2+^ homeostasis [94,95]. ERO1α and β isoforms have been described, and their distribution among tissue could be different [96]. Interestingly, unlike ERO1β, ERO1α was shown to be stimulated in an in vivo hypoxic rat model and, further, in mouse and human culture cells [96]. Conversely, the ER stressor inductor, tunicamycin, was shown to evoke an unfolded protein response (UPR) that upregulated ERO-1β, but not ERO-1α [96], thereby demonstrating two different regulatory mechanisms for each ERO1 subtype in response to different ER stressors. Regarding the physiological relevance of ERO1, it has been shown that VEGF activation, and the subsequent angiogenesis, seems to be regulated by ERO1-α downstream of HIF-1α, in response to hypoxia [97].

Histone lysine demethylases (KDMs) have been shown to be upregulated by HIF-1α due to hypoxia, like KDM3A, KDM4B, KDM4C and KDM6B [98]. The physiological relevance of this signaling pathway is based on the fact that DNA and histone methylation are the main epigenetic mechanisms for regulating gene expression. Thus, histone methylation is regulated by the balance between histone methyltransferases and histone demethylases; therefore, the activation of these proteins promotes gene expression by removing the methylation-induced break to gene translation. Interestingly, KDM3A and KDM4B genes contain three putative hypoxia response elements, which confer to this protein a unique regulatory mechanism that is driven by hypoxia as compared to other KDM family members who lack these domains [99]. In fact, the expression of KDM3A and KDM4B, but not other KDMs, was reported in U2OS, MCF7, HeLa, IMR32 and HL60 cells cultured at 0.5% O_2_ pressure [87,100].

PDK-1 and Glut-1 cancer cells exposed to hypoxic conditions often reprogram their metabolism to a pure glycolytic metabolism, which may work even in anoxic conditions. As result of this adaptation, PDK-1 and Glut-1 activities are exacerbated in order to impair pyruvate processing in the mitochondria, enhancing the glucose transport and subsequently triggering an exacerbated glycolysis. It has been shown that these metabolic changes may be supported by the mTORC1-HIF-1 axes in CD8+ T cells [101]. An early observation concluded that HIF-1 activation favored selective GLUT regulation [88,102]; thus, while low-affinity/high capacity GLUT2 activity resulted in repression, other high-affinity/low capacity isoforms, like GLUT1 and GLUT3, were upregulated during hypoxia [102]. Similarly, other glycolytic enzymes may also be overexpressed in hypoxic conditions, as with PFK-1, aldolase A, PKM, triose-phosphate isomerase and LDH-A. However, increased glycolysis would not be sufficient under conditions where cells still consume O_2_ in the tricarboxylic acid cycle within the mitochondria as a result of pyruvate degradation. Therefore, cells subjected to hypoxia often block the mitochondrial tricarboxylic cycle by enhancing phosphorylation of the pyruvate dehydrogenase due to the activation of PDK-1 [103]. Both metabolic regulatory mechanisms were observed in several cancer cells.

Osteopontin (OPN) was initially described as a regulator of osteoclast function during bone remodeling, since it was secreted and found complexed in hydroxyapatite deposits within the mineralized bone matrix. The vitronectin receptor on the surfaces of osteoclasts recognize OPN and directs these cells to the bone locations that must be remodeled [104]. Additionally, the cytokine-related role of OPN was also shown to promote interferon-gamma and interleukin-12 generation. Overexpression and upregulation of OPN by hypoxia and radiotherapy were reported in certain tumors [105]. Interestingly, in breast cancer biopsies, a positive feedback loop between hypoxia and OPN was demonstrated. The authors claim that OPN favors HIF-1 induction during hypoxia, but not HIF-2, and that subsequently, VEGF activation leads to angiogenesis. This signaling pathway would require the activation of integrin-linked kinase (ILK)/AKT-mediated nuclear factor (NF)-κB p65 [106]. The latest studies consider OPN as a good cancer molecular marker of poor prognosis and a potential plasma marker in patients suffering from different types of cancers [107,108,109]. 

BNIP-3. HIF-1α stabilization is critical for cell survival under hypoxia. However, a close look at the two transactivation domains, (N-TAD and C-TAD) of HIF-1, reveals a dual function of HIF-1, which was revealed as a result of the use of the C-TAD inhibitor [110]. Under these experimental conditions, the N-TAD-targeted genes included the Bcl-2/adenovirus E1B 19-kDa interacting protein 3 (*bnip3*) among others, the maximum expression of which resulted in severe hypoxia in areas close to the necrotic parts of the solid tumors [110]. BNIP-3 belongs to the BH3-only subfamily of the Bcl-2 protein family that, upon dimerization, antagonized the pro-survival activity of proteins such as Bcl-2 and Bcl-X_L_ [111]. It has been demonstrated that the activation of BNIP-3 evokes cell death [112]; however, exogenous overexpression of this protein in several cultured cell types, such as MEFs, MCf7, PC3 and LS174 cells, failed to reproduce previous results found in cardiomyocytes [110]. Interestingly, BNIP3 was reported to be involved in mitochondrial autophagy (mitophagy), a mechanism that may play a protective role against hypoxia-induced cell death due to the disruption of those altered mitochondria that release elevated amounts of ROS [113]. Later on, it was shown that in nucleus pulposus cells locatedwithin avascularized areas within the intervertebral discs that are constantly supporting hypoxic conditions, HIF-1 should be activated and that, subsequently, autophagy must be constantly modulated [114]. Interestingly, these cells respond to hypoxia by increasing the LC3 content, an autophagy cell marker that is involved in the generation of autophagosomes that were HIF-1/BNIP3 independent [114]. In addition, some cancer cell types overcome the hypoxia-evoked cell death by inducing aberrant methylation and the silencing of BNIP3, as was observed in colorectal and gastric cancers [115]. 

All these HRE elements are now being subjected to intense investigations in order to ascertain the hypoxia-related effect in cell physiology and in pathologic conditions; thus, the knowledge regarding these HRE will increase in the following years. Interestingly, most of these molecular markers of hypoxia have been described in platelets, with the exception of BNIP-3 [63,64,65,66]. Nonetheless, GLUT1 and HSP70 are currently being investigated in platelet. HSP70 is particularly relevant in diabetic patients [116], who have also been also described as suffering ROS overproduction and altered glucose metabolism; hence, both mechanisms seem to be highly intricate in platelets.

## 5. Platelet Function Alteration in Response to Hypoxia: HIF-1 in Platelet Function 

Wide bodies of evidence regarding the hypoxia effects on platelet function have arisen from studies conducted on subjects exposed to high altitudes Thrombotic events derived from hypoxemic conditions at high altitudes rely on the generation of a pro-thrombotic-like status, which, according to the literature, is similar to that found in patients suffering from venous thromboembolism [117]. Two recent publications detailed the molecular alterations that take place in platelets exposed to high altitudes and hypoxia. The authors concluded that, in addition to an enhanced HIF-1α expression, platelets also presented alterations in many other molecular pathways and proteins, such as elevated vWF, CD40, P-seletin and PF4, which facilitate the appearance of the coagulation and thrombophilia profiles characteristic of these patients [117]. 

These results were later corroborated by other studies using hypobaric and hypoxic conditions which demonstrated alterations in several platelet proteins, including the membrane glycoproteins, GP4, GP6 and GP9; integrin subunits (ITGA2B); and chemokines located in alpha-granules (SELP, PF4V1) [118]. In line with this, a previous publication demonstrated alterations in HIF-1α expression in subjects living at high altitude in comparison to subjects living at sea-level; another 137 genes were also differentially expressed under these conditions. It is worth mentioning that among these, the authors clearly differentiated between proteins belonging to the oxidative stress-related protein, focal adhesion, and those proteins involved in the complement and coagulation cascades, respectively [119]. Considering the latter, the authors reported an altered expression of F11, HRG, SERPINF2, SERPIND1, the expression of which was reduced at high altitudes. Meanwhile, PF4 and vWF were overexpressed. These molecular alterations resulted in a significant increase in prothrombin time and elevated partial thromboplastin time [119].

Platelet exposition to hypoxia activates HRPs; therefore, it is likely that HIF-1 regulates platelet function. Surprisingly, not many papers can be found in the literature regarding the role of HIF-1 on hypoxia-dependent alterations in coagulation; few studies directly analyze the role of HIF-1 in platelet function. This contradiction is due to cancer cells and endothelial cells, particularly those located in the lung vasculature, activates HIF-1 under hypoxic conditions, which in turn activates the secretion of PAI-1, therefore recruiting platelets to the tumor location. This aberrant platelet function avoids the detection of cancer cells by the immune system [120]. Therefore, those studies focus the interested of researches that analyze the molecules and alteration derived or affecting to endothelial cells, meanwhile those affecting directly to platelets remains elusive.

A deep search in the literature about HIF-1 role in platelets leads us to conclude that this protein participates in platelet generation and function, as it is following described. Regarding the role of hypoxia in platelet generation, it has been reported that both HIF-1 and HIF-2 were detected in megakaryocytes. In line with this observation, those patients diagnosed of immune thrombocytopenia (ITP) that are suffering from a high rate of platelet destruction, presented low HIF-1α expression in the bone marrow [121]. Additionally, administration of HIF-1α activator, IOX-2, restored both megakaryocyte maturation and platelet counts in a murine model of ITP [121]. Other authors have reported that iron deficiency leads to pathologic excessive number of platelets known as thrombocytosis, which occurred independently of the megakaryopoietic growth factors trombopoietin, IL-5 or IL-11 [122]. Interestingly, iron deficiency in cord blood cell cultures evoked the expression of megakaryopoietic markers that lead to pro-platelet formation and subsequent molecular analysis revealed enhanced HIF-2α and VEGFA. These data indicate that HIF-2 role is relevant to platelet production in response to iron deficiency [122].

Further evidence can be found in patients suffering the Chuvash polycythemia disease. These patients are homozygous for C598T mutation in the von Hippel-Lindau protein (VHL) that as mentioned above is the protein responsible for HIF-1α ubiquitination [123]. Chuvash polycythemia patients often die of thrombotic events. Authors claimed that these patients have elevated amounts of reduced glutathione (GSH), a natural cell antioxidant found in these patients that resulted in elevated glutamate cysteine ligase (GCL). In line with this experiments done using a murine model expressing VHL mutants, resulted in a decreased HIF-1 expression that positively correlates with GCL levels, then HIF-1α may impair the cellular balance of reactive oxygen species by altering the GSH generation [123]. GHS has been described to contribute to the appearance of vascular diseases due to increasing platelet function [124], in fact, GHS intracellular production in response to platelet agonists like Thr or arachidonic acid, is involved in thromboxane B_2_ and, subsequently, contributes with platelets aggregation [125].

Finally, later evidence regarding the role of hypoxia in platelet function turn up of the increased number of patients suffering sleep apnea and chronic obstructive pulmonary diseases [43]. In normoxic conditions circulating platelets express HIF-2α that is exacerbated in hypoxic conditions associated with chronic obstructive pulmonary disease, while mRNA of HIF-1α and HIF-2α was found in circulating platelets [43]. Thrombin and other platelet agonists also evoke increasing HIF-2α expression which leads to PAI-1 expression and secretion that in turn activates platelets, closing the hypoxia platelet activation cycle [43]. Recently, it has been published a list of microRNA associated to hypoxia and coagulation after an in silico extensive study [126]. Authors claimed that hsa-mir-4433a-3p, hsa-mir-4667-5p, hsa-mir-6735-5p, hsa-mir-6777-3p and hsa-mir-6815-3p regulates simultaneously hypoxia related genes (among them HIF-1α, HIF-2α and HIF-3α, Arnt and Arnt2) and genes involved in coagulation, which may be the start point for interesting future investigations, but without the appropriate experimental confirmation these data lack of physiological relevance. Finally, HIF prolyl-hydrolases inhibition using IOX-2, a dual PHD and HIF antagonist, was evidenced to impair platelet aggregation in response to CRP and thrombin [19]. Authors claimed that HIF-1α expression was enhanced in platelets due to the antagonist treatment that is also linked with a reduction in ROS production [19]. 

Finally, as shown in Figure 2, HIF-1α plays a crucial role in chronic hypoxia by inducing the TRPC-SOCE signaling axis. Numerous studies from different groups have reported on the overexpression of, for example, TRPC6 and TRPC1 in pulmonary arterial smooth muscle cell (PASMC) [127]; RPC3, TRPC6 and TRPC1 in cultured neonatal rat cardiac myocytes [128]; and TRPC6 with enhanced HIF-1α/ZEB2 axis in a middle cerebral artery occlusion (MCAO) model [129]. Nonetheless, there is not a unique association between HIF-1α and SOCE members. HIF-1α directly controls STIM1 transcription and is required for STIM1-mediated SOCE via the activation of Ca^2+^/calmodulin-dependent protein kinase II and p300 in hepatocarcinoma cells [130]. In ovary carcinoma cells exposed to placental growth factor (PlGF), Orai1/STIM1 expression is enhanced, contributing to the upregulation of HIF-1α [131]. Conversely, Wang et al., 2017 reported HIF-1α-dependent upregulation of Orai2, but not Orai1, in pulmonary arterial smooth muscle cells (PASMCs) and mice models [132]. Orai3 is induced by hypoxic environments in some cancer cells, for example, the induction of Orai3 under hypoxia is mediated by HIF-1α in the TNBC MDA-MB-468 cell line, increasing the TRPC1 ion channel induced by HIF-1α [133]. Curiously, in spite of the fact that TRPC1 silencing suppresses HIF-1α induction by hypoxia, Orai3 does not show the same results [134].

## 6. Conclusions

Platelet function is modified during perinatal hypoxia. The main HRPs, HIF-1 and HIF-2, were reported to regulate platelet production and function, as further confirmed by the deregulated platelet function shown in patients suffering from Chuvash polycythemia disease. Hypoxia modifies, among others, HIF-1 function, which might in turn modify Ca^2+^ homeostasis in platelets, leading to altered platelets function during labor. Therefore, further research is required to demonstrate the presence of the hypoxia-regulated mechanism discussed in this review. 

## Figures and Tables

**Figure 1 ijms-24-14541-f001:**
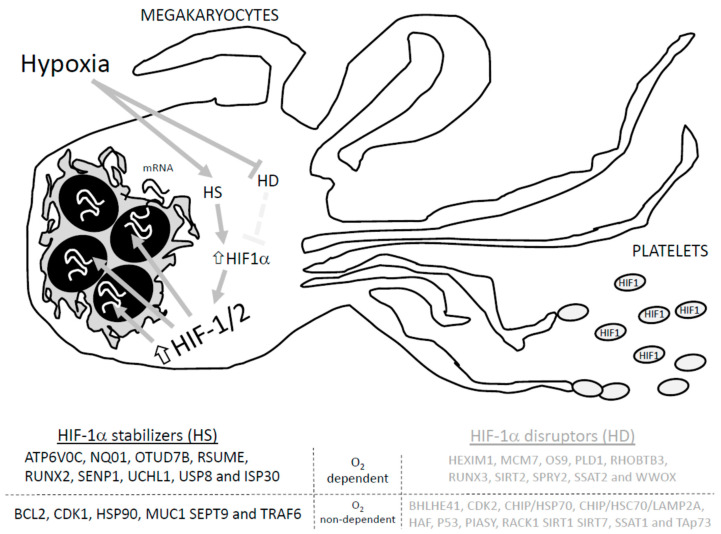
Proteins involved in HIF-1a stabilization and degradation. Adapted from Semanza GL, 201855. HD: HIF-1 disruptors; HS: HIF-1 stabilizers. Hypoxia activates hypoxia stabilizers (HS) while blocking hypoxia disrupters (HD), which increases HIF-1α conforming to the HIF-1/2 complex that induces translation of the targeted genes, and promotes the arresting of HIF1 in the generated platelets.

**Figure 2 ijms-24-14541-f002:**
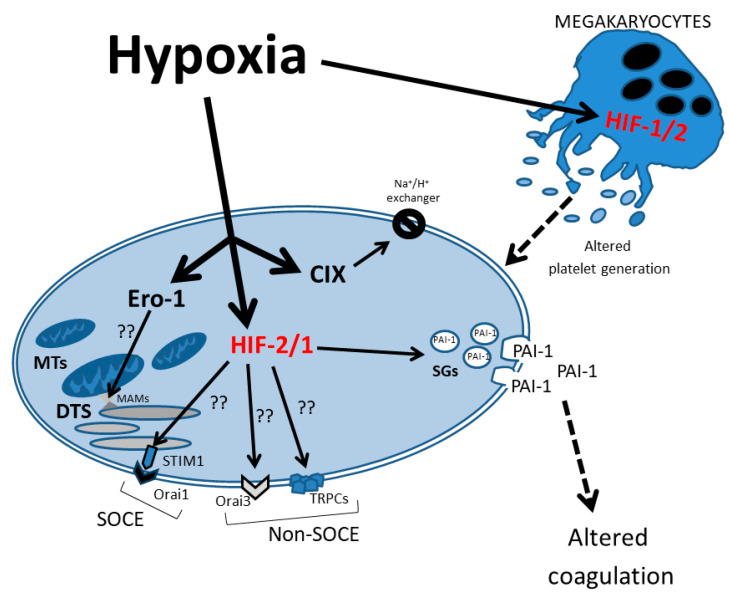
Schematic representation of hypoxia effects on the signaling pathways in neonatal platelets. The HIF-1/2 regulatory effect on several proteins that control intracellular Ca^2+^ homeostasis is represented. DTS: dense tubular system; MTs: mitochondria; SGs: secretory granules; PAI-1: plasminogen activator inhibitor 1; SOCE: store-operated calcium entry; MAMs: mitochondria-associated membranes.

**Table 1 ijms-24-14541-t001:** Hypoxic-response proteins (HRPs) with a possible role in platelet physiology.

Hypoxia Marker	Platelets *	Expression in Other Cells	References
KDM4B & KDM3A	Yes	U2OS, MCF7, HeLa, IMR32 and HL60 cell lines	Van Oorschot RV et al. [87]
GLUT1	Yes	Universally expressed	Filder TP et al. [88]
Osteopontin	Yes	Widely expressed	Chanzu et al. [89]
BNIP-3	No	Megakaryocytes, other stem cells, breast cancer cells, etc.	Malherbe JAJ et al. [90]
HSP-70	Yes	Caco-2 & HT29 cell lines	Jackson et al. [91]

* Corroborated at protein level in platelets.

## Data Availability

Not applicable.

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
