# Peer review of "Thrombotic Alterations under Perinatal Hypoxic Conditions: HIF and Other Hypoxic Markers"

_ijms, 2023, doi:10.3390/ijms241914541_

Round 1

Reviewer 1 Report

Very nice and good written review, however has no real connection with regulation of platelet functions under hypoxic conditions. Therefore, I will suggest that the authors should change the title of the review (to not include platelets), or add literature directly related to platelet function during hypoxia. In PubMed “platelet hypoxia” shows more than 2600 papers, of course not all of them directly connected with platelet function under hypoxic conditions, however it is extensively studied especially at high altitude and there are still no consensus in the literature whether hypoxia is connected with platelet inhibition or activation. If the authors want to keep platelet in the title of this review, they should include above-mentioned literature and critically analyzed it.

Below are listed some comments that the authors should correct.

1.      P. 5, L. 240-243. The sentence “Therefore, vasoconstrictors activate G-protein-coupled receptors, increase inositol triphosphate (IP3) and inhibits the production of cyclic guanosine monophosphate (cGMP) in SMCs and enhanced SOCE; thus, promoting their contraction and the maintenance of PHN.” should be corrected. Which GPCRs, and activation of GPCRs are not connetde only with IP3, it is too primitive and should be written more accurate. Abbreviation of SOCE was already mentioned at p. 4, L. 201-202

2.      “Table I. Hypoxic-response proteins (HRPs) with possible role in the platelet physiology.”  In this table presented the literature, but it was not analyzed in the text.

3.      P. 11, L. 498-499. “GHS has been described to contribute to the appearance of vascular 498 diseases due to increasing platelet function [117].” What means “increasing platelet function”? Platelet activation?

4.      HIF-1 and HIF-2 expression in platelets. P. 11, L. 503. “mRNA of HIF-1α and HIF-2α was found in circulating platelets[40].” In this paper only HIF-2α is shown in platelets.   

Author Response

REVIEWER 1

Very nice and good written review, however has no real connection with regulation of platelet functions under hypoxic conditions. Therefore, I will suggest that the authors should change the title of the review (to not include platelets), or add literature directly related to platelet function during hypoxia. In PubMed “platelet hypoxia” shows more than 2600 papers, of course not all of them directly connected with platelet function under hypoxic conditions, however it is extensively studied especially at high altitude and there are still no consensus in the literature whether hypoxia is connected with platelet inhibition or activation. If the authors want to keep platelet in the title of this review, they should include above-mentioned literature and critically analyzed it.

Answer: We would like to thank the nice and constructive comments done by the reviewer regarding our manuscript. Due to the actual review is already very extensive in its present form, we have briefly introduced the reviewer’s suggestion regarding the effects of the hypoxia in activation of platelets. Thus, we have focused in some references that explain the changes in platelet function at high altitude where often exits low oxygen supply and hypoxemia conditions may appear. Additionally, “platelet” has been removed from the title as suggested.

Below are listed some comments that the authors should correct.

  1. 1. 5, L. 240-243. The sentence “Therefore, vasoconstrictors activate G-protein-coupled receptors, increase inositol triphosphate (IP3) and inhibits the production of cyclic guanosine monophosphate (cGMP) in SMCs and enhanced SOCE; thus, promoting their contraction and the maintenance of PHN.” should be corrected. Which GPCRs, and activation of GPCRs are not connetde only with IP3, it is too primitive and should be written more accurate.

Answer: As suggested by the reviewer, we have described this mechanism in more detail. We agree with the reviewer that there are additional pathways activated in adult smooth muscle cells exposed to hypoxia, but we have focused in the data available that report changes in neonates SMC, and assuming that to extrapolate this data from adults may be misleading.

     Abbreviation of SOCE was already mentioned at p. 4, L. 201-202

     Answer: We have now corrected this mistake in the text and SOCE is only explained the first time that appears in the text.

  1. “Table I. Hypoxic-response proteins (HRPs) with possible role in the platelet physiology.” In this table presented the literature, but it was not analyzed in the text.

Answer: In our previous version we focused in the role of HIF-1 in platelets, but as suggested by the reviewer, we have included a new paragraph to briefly develop the literature mentioned in Table 1. But among those references, only GLUT1 and HSP70 (HSP90) have clearly stablished a relevant role of these proteins in the platelet function.

  1. P. 11, L. 498-499. “GHS has been described to contribute to the appearance of vascular 498 diseases due to increasing platelet function [117].” What means “increasing platelet function”? Platelet activation?

Answer: We indicated the increase in platelet activation leading to thrombotic events. This sentence has been modified to clarify this concept. Although effects of GHS in platelets may be contradictory depending of the agonist used, we have included a new reference that demonstrate that during Thr- and arachidonic acid-evoked platelet activation, GHS is required during thromboxane B2 generation and subsequently, for platelet aggregation (Burch JW et al., Prostaglandins 1990). Hence, the mechanisms that promote GHS generation would evoke increased platelets activation “perse”.

  1. HIF-1 and HIF-2 expression in platelets. P. 11, L. 503. “mRNA of HIF-1α and HIF-2α was found in circulating platelets[40].” In this paper only HIF-2α is shown in platelets.

Answer: We agree that the mentioned manuscript do not confirmed the expression HIF-1 at protein levels in platelets, but they have confirmed the presence of mRNA, which under certain circumstances may facilitate HIF-1 platelets expression. Conversely, a recent paper claims that HIF-1 prolyl hydrolases inhibition enhanced HIF-1a expression and, subsequently, altering platelet function. Hence, HIF-1 has to be considered as a HRP in platelets from now on. The new reference has been also included in the text (Gu W et al Trombosis and Haemostasis 2022).

Reviewer 2 Report

This manuscript is an extensive Review of hypoxia in fetuses and newborns, with alterations in platelet proteins resulting in severe complications. As platelet alterations in hypoxia following labor is less investigated, the manuscript could be of interest to the field. However, major revisions are needed to ensure cohesiveness to the paper:

1. Long paragraphs should be divided into sub-sections with headings. For example, in part 2. Hypoxia during labor, the first half of the section is about fetus' transient adaptive mechanisms against hypoxia. The second half is more about complications of prolonged hypoxia for babies. Please consider splitting this section into 2 smaller sections. Same for many other long paragraphs. 

2. In part 2.2. Hypoxic-evoked thrombotic alterations during labor, please introduce megakaryocytes at the beginning of this section together with platelets. The authors didn't introduce megakaryocytes until the end of the paragraph. 

3. Part 3. altered pathways during exposition of platelets to hypoxia, consider diving into sub-sections about vessel wall & endothelial cells and platelets. Pulmonary hypertension can be combined into the vessel wall & EC section. Check all other sections too. 

4. Some references in this Review are very dated. For example, [30] and [23]. Please make sure to include more current literature. 

5. Overall, the manuscript needs to be exhaustively re-organized into paragraphs that have clear topics/themes. There are many repetitive discussions all over the manuscript, which can be condensed. The authors need to be careful not to revisit key points that have been previously discussed in another section. Many paragraphs read like the authors just went on tangents, and not really focused on an idea. Each paragraph should only discuss 1 main point. 

Check spelling & grammar throughout. 

Author Response

This manuscript is an extensive Review of hypoxia in fetuses and newborns, with alterations in platelet proteins resulting in severe complications. As platelet alterations in hypoxia following labor is less investigated, the manuscript could be of interest to the field. However, major revisions are needed to ensure cohesiveness to the paper:

Asnwer: We thank the reviewer for his/her constructive comments regarding our draft.

  1. Long paragraphs should be divided into sub-sections with headings. For example, in part 2. Hypoxia during labor, the first half of the section is about fetus' transient adaptive mechanisms against hypoxia. The second half is more about complications of prolonged hypoxia for babies. Please consider splitting this section into 2 smaller sections. Same for many other long paragraphs. 

Answer: We agree that some paragraphs may be difficult to read and they have been simplified in the new version of the manuscript. Further, we have included new subheading to clearly guide the readers throughout the text.

  1. In part 2.2. Hypoxic-evoked thrombotic alterations during labor, please introduce megakaryocytes at the beginning of this section together with platelets. The authors didn't introduce megakaryocytes until the end of the paragraph. 

Answer: As suggested, we have now described the role megakaryocytes by including a brief description of the mechanism underlying platelet production from megakaryocytes, which would help the readers that are not familiar with the topic to understand the relevance of hypoxia and HIF-1 in the megakaryocytes and subsequently, their involvement in the haemostasis.

  1. Part 3. altered pathways during exposition of platelets to hypoxia, consider diving into sub-sections about vessel wall & endothelial cells and platelets. Pulmonary hypertension can be combined into the vessel wall & EC section. Check all other sections too. 

Answer: We have included subheading trying to clarify this section of the text. Thank for the suggestion done by the reviewer.

  1. Some references in this Review are very dated. For example, [30] and [23]. Please make sure to include more current literature. 

Answer: Those references may be very dated ones, but they are solid ones and have been published in relevant journals. Since they have not been rebutted by new ones yet, they are still valid references and, therefore, we have decided that they deserved to appear in the publication. Conversely, we have included some new references that reinforce those very preliminary publications as suggested.

  1. Overall, the manuscript needs to be exhaustively re-organized into paragraphs that have clear topics/themes. There are many repetitive discussions all over the manuscript, which can be condensed. The authors need to be careful not to revisit key points that have been previously discussed in another section. Many paragraphs read like the authors just went on tangents, and not really focused on an idea. Each paragraph should only discuss 1 main point. 

Answer: Thank for your comments and suggestions. We have now corrected those expressions and paragraphs that are difficult to understand, or were not enough attractive for readers due to their length. In addition, a revision of the grammatical and typo mistakes has been done. We apologize for including them in our previous draft.

Round 2

Reviewer 2 Report

I thank the authors for addressing my comments. The manuscript has been substantially reorganized and improved. 

N/A